# Active TB case finding in a high burden setting; comparison of community and facility-based strategies in Lusaka, Zambia

**Mary Kagujje**[1]*, **Lophina Chilukutu**[1], **Paul Somwe**[2], **Jacob Mutale**[2], **Kanema Chiyenu**[2], **Mwansa Lumpa**[2], **Winfrida Mwanza**[1], **Monde Muyoyeta**[1]

**1** Tuberculosis Department, Centre for Infectious Disease Research in Zambia, Lusaka, Zambia, **2** Strategic Information Department, Centre for Infectious Disease Research in Zambia, Lusaka, Zambia

* mkagujje@gmail.com

## Abstract

### Introduction

We conducted an implementation science study to increase TB case detection through a combination of interventions at health facility and community levels. We determined the impact of the study in terms of additional cases detected and notification rate and compared the yield of bacteriologically confirmed TB of facility based and community based case finding.

### Methodology

Over a period of 18 months, similar case finding activities were conducted at George health facility in Lusaka Zambia and its catchment community, an informal peri-urban settlement. Activities included awareness and demand creation activities, TB screening with digital chest x-ray or symptom screening, sputum evaluation using geneXpert MTB/RIF, TB diagnosis and linkage to treatment.

### Results

A total of 18,194 individuals were screened of which 9,846 (54.1%) were screened at the facility and 8,348 (45.9%) were screened in the community. The total number of TB cases diagnosed during the intervention period were 1,026, compared to 759 in the pre-intervention period; an additional 267 TB cases were diagnosed. Of the 563 bacteriologically confirmed TB cases diagnosed under the study, 515/563 (91.5%) and 48/563 (8.5%) were identified at the facility and in the community respectively (P<0.0001). The TB notification rate increased from 246 per 100,000 population pre-intervention to 395 per 100,000 population in the last year of the intervention.

### Conclusions

Facility active case finding was more effective in detecting TB cases than community active case finding. Strengthening health systems to appropriately identify and evaluate patients

**Data Availability Statement:** All relevant data are within the manuscript and its Supporting Information files.

**Funding:** Initials: MM Grant Number: STBP/
TBREACH/GSA/W5-26 Funder: The Stop TB
Partnership's TB REACH initiative with funding
from the government of Canada URL: http://www.
stoptb.org/global/awards/tbreach/. The funders
had no role in study design, data collection and
analysis, decision to publish, or preparation of the
manuscript.

**Competing interests:** The authors have declared
that no competing interests exist

for TB needs to be optimised in high burden settings. At a minimum, provider initiated TB symptom screening with completion of the TB screening and diagnostic cascade should be provided at the health facility in high burden settings. Community screening needs to be systematic and targeted at high risk groups and communities with access barriers.

## Introduction

Of the estimated 10 million incident TB cases globally, in 2018, only 7 million were notified [1]. Zambia, a high TB burden country, has an estimated TB treatment coverage of only 58% [1]. The country has an estimated 24,929 missing TB cases [2] and most of these cases are expected to be found in large peri-urban informal settlements of large cities [3]. Based on data from the TB prevalence survey, about 50% of the symptomatic TB cases are missed at the health facility [4]. TB cases are missed at the health facility due to low index of suspicion of TB, failure to complete the TB diagnostic cascade, use of less sensitive diagnostic tools, out of pocket expenditure for patients and weak public private coordination [4–7].

The undiagnosed and untreated TB cases are key factors contributing to the continued global TB epidemic; perpetuating TB transmission and increased risk for adverse outcomes due to delayed diagnosis [8–10]. Finding the missing TB cases is thus a global priority for TB control [11] and Active Case finding (ACF) has been identified as one of the key components to achieving this [12].

Much as the term ACF is often used to imply systematic screening and diagnostic evaluation of TB risk groups that happens outside the health facility, it actually constitutes provider initiated screening both inside and outside the health facility [13]. There is evidence on effectiveness of ACF in the community [14–19], there is less evidence for the effectiveness of ACF at the health facility [7,20,21] and even less literature comparing the two active case finding strategies [21].

An implementation science study was conducted at a primary health care facility in Lusaka district, Zambia. The objective of the study was to increase TB case detection through a combination of interventions at both the health facility and community level. Additionally, the study assessed and compared the contribution of facility based and community based ACF activities to TB case detection. We report the impact of the study on TB notification in terms of additional cases detected and notification rate and compare the yield of facility based and community based case finding.

## Methods

### Study setting and study population

This study was undertaken between July 2017 and December 2018 in a TB programmatic setting at George primary health care TB diagnostic facility and its catchment population. George community is an informal, poor, high density peri-urban settlement in Lusaka district in Zambia: Fig 1.

Lusaka province with a prevalence of 932/100,000 population, has the second highest burden of TB in Zambia after the Copper belt province [22]. The notification rate of TB in Lusaka district in 2016 (pre intervention period) was 640/100,000 (Lusaka District TB data, unpublished), above the country average of 236/100,000 population [23]. In the same year George health facility had a notification rate of 246/100,000 population. George health facility has an

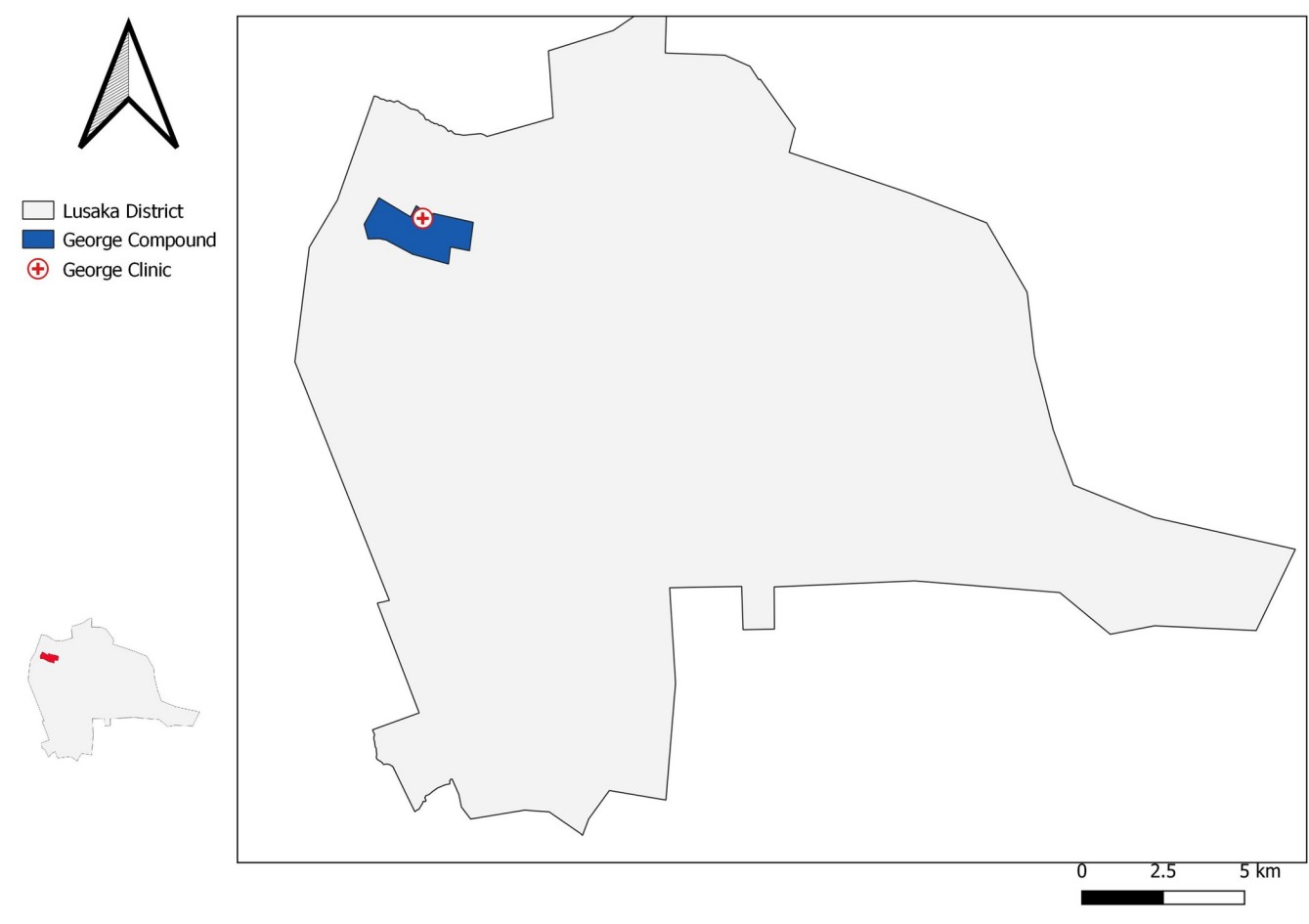

**Fig 1. Geographical location of George primary health care centre and community.**

outpatient department (OPD), antiretroviral therapy (ART) clinic, Maternal Child Health (MCH) clinic, a voluntary counselling and testing (VCT) point and TB clinic. The catchment community population was 166,975, 173,130 and 179,360 people in 2016, 2017 and 2018 respectively. Before the study, the clinic had onsite LED microscope with no onsite chest x-ray and geneXpert; a mobile digital x-ray and a geneXpert were installed during the study.

## Study procedures

Similar case finding activities were conducted at the health facility and in the community; they included awareness and demand creation activities, TB screening, diagnosis and linkage to treatment.

**Awareness and demand creation activities.**   First, we re-oriented facility health workers and trained community health workers on TB to raise their index of suspicion of the disease. At the health facility, we displayed posters on TB symptoms, community health workers provided daily health talks on TB in all the departments of the clinic and distributed flyers on TB. In the community, we provided door to door sensitization on TB, conducted drama sensitization and displayed posters in places that have/attract large numbers of people and distributed flyers on TB. All these activities had messaging encouraging people to screen for TB.

**TB screening and diagnosis.**   At the health facility, a trained community health worker was stationed at each department to register patient details and refer patients for X-ray

screening. In addition, an open access point manned by community health workers was set up to provide fast track TB screening and diagnostic evaluation for clients that were referred by the clinicians and the community health workers and clients presenting directly from the community. In the community, screening and sputum collection points were set up in each mapped zone and identified congregate settings in a rolling fashion with repeated rounds to ensure saturation.

History of the four World Health Organisation (WHO) recommended symptoms for TB screening (cough, fever, night sweats and weight loss) [13] and 2 additional symptoms from the Zambia TB guidelines (chest pain and loss of appetite) [24] was documented for all patients presenting for TB screening. One mobile digital chest x-ray (CXR) from Delft Imaging Systems with Computer Aided Diagnosis (CAD4TB) version 5 was used both for community and facility TB screening. Two WHO recommended algorithms [13], both similar to the standard of care algorithms in Zambia except for duration of symptoms when symptom screening is used [24] were used to evaluate for TB: 1) When CXR was available, all patients were screened with CXR-CAD4TB irrespective of symptoms followed by Xpert for those with abnormal CXR; abnormal CXR was defined as CAD score above 60 and 2) When CXR was not available, individuals with any of the above symptoms, irrespective of duration submitted a sputum sample for Xpert. Additionally, clinicians had the discretion to request for GeneXpert for patients who were symptomatic but with a CAD score below 60.

Each patient was instructed on how to collect a quality sputum sample by a community health worker. All samples were triple packaged before transportation to the laboratory by community health workers on the same day of collection. Samples were rejected by the laboratory if: i) the specimen was leaking out into biohazard bag, ii) the sputum contained many food particles, iii) the volume was less than <0.5mls and if the sputum contained a lot of blood.

HIV status was either self-reported or obtained through opt out HIV testing.

All patients diagnosed with TB that did not return to the screening point for results within 2 days had a home visit carried out by a community health worker to facilitate linkage. Contact tracing was done for TB cases identified during the study per routine service requirements.

## Data collection and data management

Data was collected from the study TB screening registers and the existing approved National TB laboratory register, TB treatment register and household contact register. The study TB community and facility screening registers were a modification of the nationally approved presumptive TB register whose additional data elements included history of TB treatment, history of contact to a TB case, duration of cough and CAD score. Data from contact tracing was reported under community screening. Data from the facility and community screening registers was entered into a customized web application operating with a Microsoft SQL Server database backend. Transact SQL queries were used to generate weekly/biweekly reports. Error reports were used to flag data inconsistencies that needed corrective actions to be taken ensuring data integrity. Incremental database backups were made on a daily basis.

Data comparing community and facility case finding was obtained from the screening registers while data on impact of the interventions in terms of additional cases and notification rate was obtained from the facility TB treatment register.

## Data analysis

Data was analysed using STATA Statistical Software (Stata Corporation Version 14. College Station, Texas 77845, USA). To show the flow of patients through the diagnostic cascade, 2

flow diagrams were generated for facility and community based case finding and each showed the following steps: individuals screened with presumptive TB, individuals who submitted a sputum sample, individuals with sputum sample evaluated and individuals with sputum evaluated who were diagnosed with bacteriologically confirmed TB(yield). To determine any facility level and community level population characteristic differences among those screened that might account for the differences in case detection, a 2X2 table was constructed and categorical variables were compared using the Chi-squared test and continuous variables using the student t-test.

Additional analysis was done to determine the contribution from the community and facility to the total cases detected; contribution from facility was disaggregated further by entry point to determine which entry point had the highest yield.

To determine the impact of the case finding on TB notifications, additional cases detected were calculated by comparing TB notifications during the intervention period to a corresponding pre-intervention period. The intervention period included notification data from 3rd quarter 2017 to 4th quarter 2018 and the pre- intervention period included notification data from 3rd quarter 2015 to 4th quarter 2016. Additionality was the difference in TB notification between the intervention period and the pre-intervention period and the percentage change was the additional cases divided by the total notifications in the pre-intervention period multiplied by 100 percent. Lastly, changes in notification rates were also determined taking into consideration the catchment population.

### Ethical issues

Approval to conduct the study was provided by the University of Zambia Biomedical Ethics Research Committee (UNZA BREC) No: 012-05-17 and National Health Research Authority. A waiver of written consent was given by UNZA BREC as the study operations were routine. However, verbal consent was given before participation in the study.

### Results

A total of 18,194 individuals were screened for TB under the study; 9,846(54%) were screened at the facility while 8,348(48%) were screened in the community. The characteristics of patients screened for TB in the facility and community are illustrated in Table 1. There were 5,053/9,846(51.3%) males among the individuals screened at the facility and 4,256/8348 (51.0%) males among the people screened in the community (P = 0.588). The mean age of individuals screened in the facility was 35.2 (SD 14.2) while in the community it was 31.3 (SD

**Table 1. Description and comparison of facility and community patients.**

| Characteristic | Facility 9,846 (%) | Community 8,348 (%) | P-Value |
|---|---|---|---|
| Male sex | 5,053 (51.3) | 4,256 (51.0%) | 0.588 |
| Mean age (sd) | 35.2 (14.2) | 31.3 (15.5) | <0.0001 |
| HIV positive status | 4,183 (42.5) | 718 (8.6) | <0.0001 |
| Previous TB | 1,462 (14.8) | 474 (5.7) | <0.0001 |
| **Symptoms** * | | | **<0.0001** |
| No symptoms | 1,864 (18.9) | 3, 108 (37.2) | |
| 1 symptom only | 2,134 (21.7) | 2,261 (27.1) | |
| 2 or more symptoms | 5,573 (56.7) | 2,824 (33.8) | |
| Abnormal CXR | 818 (13.8) | 229 (4.7) | <0.0001 |

*missing symptoms facility = 275, community = 155

15.5) (P <0.0001). Among individuals screened at the facility, 4184 /9846 (42.5%) were HIV positive while 718/8348 (8.6%) of those screened in community were HIV positive (P <0.0001). History of previous TB was 1,462/9846(14.8%) at the facility and 474/8348(5.7%) in the community (P <0.0001). Of individuals screened at the facility, 1,864/9846(18.9%) were asymptomatic (had none of the 6 symptoms used for TB screening) while 3,108/8348(37.2%) of those in the community were asymptomatic (P <0.0001).

Of the individuals screened at the health facility, 6403/9846(65%) met the definition of presumptive TB, of which 5701/6403 (89%) submitted sputum for evaluation, 3528/5701 (62%) sputum samples were evaluated, and 506/3528(14.3%) had bacteriologically confirmed TB. An additional 220 sputum samples were collected from patients who didn't meet the definition of presumptive TB. Of these, 9/220(4%) had bacteriologically confirmed TB. The total number of bacteriologically confirmed TB cases at the facility was 515 as illustrated in Fig 2. The overall yield for facility case finding was 515/3748 (13.7%)

Of the individuals screened for TB in the community, 2531/8358(30%) met the definition of presumptive TB, of which 1295/2531 (51%) submitted sputum for evaluation, 1165/1295 (90%) sputum samples were evaluated, and 42/1165(3.6%) had bacteriologically confirmed TB. An additional 404 samples were collected from individuals who didn't meet the definition of presumptive TB. Of these, 6/404(1%) had bacteriologically confirmed TB. The total number of bacteriologically confirmed TB cases was 48 as illustrated in Fig 3. The overall yield for community case finding was 48/1569 (3.1%)

The total number of bacteriologically confirmed TB cases detected was 563. Of these 515/ 563 (91.5%) were detected at the facility and 48/563 (8.5%) were detected in the community. At the health facility, 49/515(8.7%) TB cases were from ART clinic, 3/515(0.5%) TB cases were from MCH, 232/515(41.2%) TB cases were from OPD, 214/515(38%) TB cases from the fast track, 2/515(0.4%) TB cases were from TB clinic and 9/515(1.6%) TB cases were from VCT: Table 2.

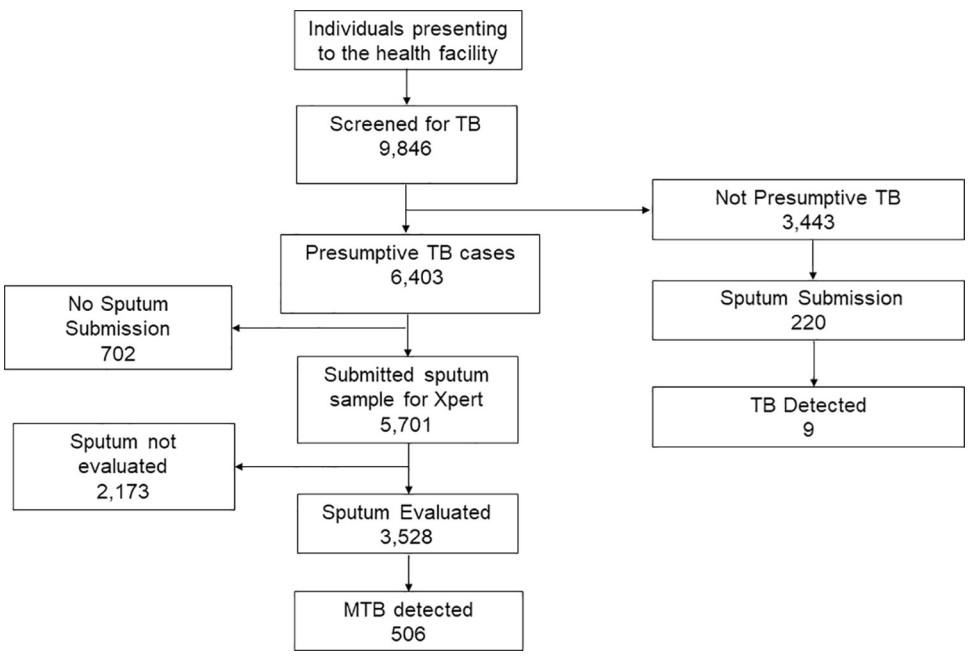

**Fig 2. Flow diagram of individuals screened at facility level.**

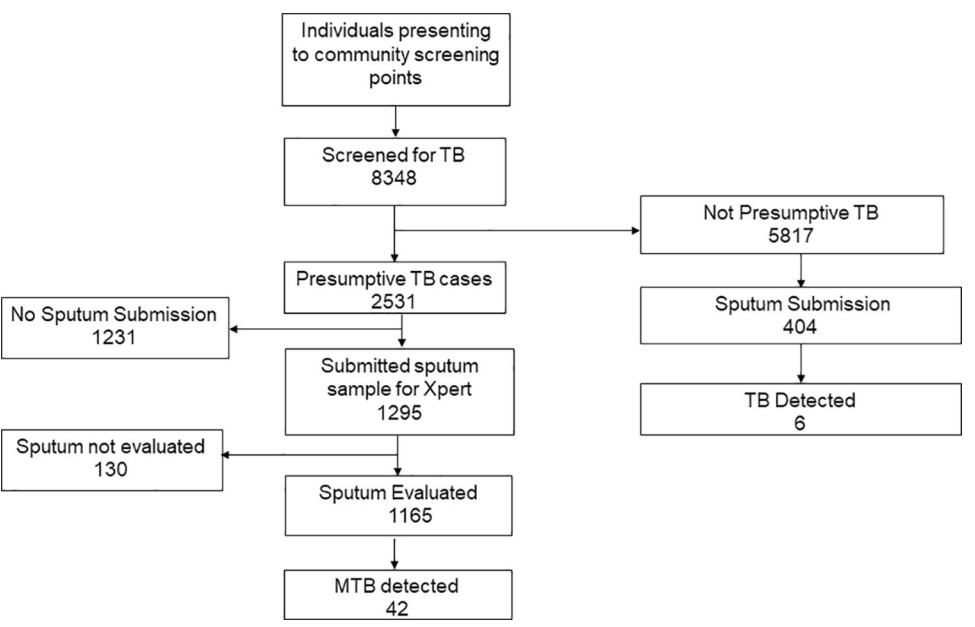

**Fig 3. Flow diagram of individuals screened at community level.**

A comparison of the TB notifications before the intervention and during the intervention as per the TB treatment register is shown in Table 3.

During the period 18 months before the intervention, 759 TB cases were notified with 272/759(35.8%) of the TB cases being bacteriologically confirmed pulmonary cases, 324/759 (42.6%) being clinically diagnosed pulmonary cases and 163/759(21.4%) being extra pulmonary TB cases: Table 4. In the 18 months of the intervention, 1026 TB cases were notified with 598/1026(58.3%) of the TB cases being bacteriologically confirmed pulmonary cases, 361/1026 (35.2%) being clinically diagnosed pulmonary cases and 67/1026(6.5%) being extra pulmonary TB cases: Table 4

The TB notification rate changed from 247 per 100,000 population in 2016, pre-intervention to 310 per 100,000 population in 2017 during which the intervention started in July, to 394 per 100,000 in 2018 during which the intervention span the entire year: Table 5

## Discussion

An additional 267 TB cases were found during the intervention period and there was a significant increase in notification rate; these cannot be credited to a single strategy but rather a

**Table 2. TB cases detected by screening entry point.**

| Area of ACF activity | Bacteriologically confirmed TB 563 (%) |
|---|---|
| **Community** | 48 (8.5) |
| **Overall facility** | 515 (91.5) |
| ART | 49 (8.7) |
| MCH | 3 (0.5) |
| OPD | 232 (41.2) |
| Fast track | 214(38.0) |
| TB Clinic | 2 (0.4) |
| VCT | 9 (1.6) |

**Table 3. Notifications per quarter.**

| Period | Quarter | Total notifications |
|---|---|---|
| **Pre-intervention** | 2015 Q3 | 159 |
| | 2015 Q4 | 188 |
| | 2016 Q1 | 127 |
| | 2016 Q2 | 88 |
| | 2016 Q3 | 87 |
| | 2016 Q4 | 110 |
| **Intervention** | 2017 Q3 | 182 |
| | 2017 Q4 | 137 |
| | 2018 Q1 | 177 |
| | 2018 Q2 | 178 |
| | 2018 Q3 | 194 |
| | 2018 Q4 | 158 |

combination of strategies including awareness and demand creation activities, increased index of suspicion of TB, increased access to TB screening and diagnostic tools and use of more sensitive screening and diagnostic tools. However, this additionality is also suggestive that TB cases were previously missed, especially at the facility. This is consistent with the findings from the Zambia TB prevalence survey [4] and studies done in other settings [25–27] and calls for urgently strengthening health systems so that TB cases who present to the health facility are not missed.

In terms of case finding at facility and community level, the data from Table 1 shows that patients screened in the facility are different from patients screened from the general community; patients from the health facility are more likely to be symptomatic and have risk factors for TB and this is expected. This explains why the yield from facility-based case finding is higher than community based screening. In fact, the facility yield exceed the 10% target recommended in the National TB guidelines on facility case finding [24]. It is interesting to note that screening of patients from OPD gave the highest yield, even higher than ART department that has people at increased risk of TB. This calls for regular screening of patients presenting to OPD for TB and ensuring improved infection control practices in these settings that are often overcrowded. The high yield from fast track could be suggestive that long waiting time at the health facility could be a barrier to TB diagnosis and that fast track services potentially bypass this barrier. Overcrowded, busy facilities should consider using fast track TB services. These should be placed in a visible easy to access part of the clinic and should use the most efficient triage system to minimise waiting times.

Available literature on community TB case finding provides evidence for its effectiveness, however, also shows a low yield from the intervention. In a study done in rural South Africa comparing the yield of facility based screening for TB and contact tracing, which targets a high

**Table 4. Comparison of before and after notification by type of TB.**

| Type of TB | Notification pre-intervention (Q3 2015-Q4 2016) | Notifications during intervention (Q3 2017-Q4-2018) | Change (%) |
|---|---|---|---|
| Pulmonary bacteriologically confirmed TB | 272 | 598 | 326(120%) |
| Pulmonary clinically diagnosed TB | 324 | 361 | 37(11%) |
| Extra pulmonary TB | 163 | 67 | -96(-59%) |
| Total | 759 | 1,026 | 267 (35%) |

**Table 5. Comparison of before and after notification by type of TB.**

| Year | 2016 | 2017 | 2018 |
|---|---|---|---|
| Population | 166,975 | 173,130 | 179,360 |
| Total notifications | 412 | 536 | 707 |
| Notification rate (per 100,000 population) | 246 | 310 | 394 |

risk group for TB in the community, facility based screening yielded 11% more cases [21]. Low yield from community-wide TB screening is also seen in a study done in Vietnam where only 94 bacteriologically confirmed TB cases were detected over a 4 year period [17] and a study done in Combodia where 315,872 individuals were screened for TB to identify 783 TB cases [14]. These findings suggest that community activities for TB should be focused on awareness raising and demand creation with referral systems for TB evaluation at facility level for those that need to be evaluated. Community based collection of sputum samples should be limited to hard to reach areas that have limited access to health services.

There were 15 patients that were neither symptomatic for TB nor had abnormal x-ray but were diagnosed for TB. This points to the sensitivity gaps of both symptom screening and chest x-ray screening. There is need to profile these patients so as to provide additional lessons for future TB case finding activities.

There was a significant number of presumptive TB cases who didn't provide sputum, that is, 11% at the facility and 49% from the community. These could possibly be asymptomatic individuals with abnormal x-ray who couldn't expectorate sputum at the time. Also, the number of rejected samples was high at 2303/6996(33%) suggesting that the study could have done more in instruction of patients on collection of quality sputum samples. However, these forms of attrition are common in active TB case finding studies. In the community wide screening study in Vietnam, an average of 70% of presumptive TB cases submitted sputum and only about 40% of the sputum samples were evaluated [17]. In another active case finding study in India, only 54% of the presumptive TB cases had their sputum evaluated [28].

This study used evidence based recommendations that are already incorporated into the National TB guidelines and national strategic plans of several high burden countries [29–32] so it is easy to replicate in various settings. The weakness of this study is that it had little focus on children, a population vulnerable to TB.

## Conclusions

Overall, active case finding increases TB case detection. In this high burden TB setting, facility based active case finding was significantly more effective than community based active case finding. Strengthening health systems to appropriately identify and evaluate patients for TB needs to be optimised in high burden settings with low TB case detection rates. At a minimum, provider initiated TB symptom screening with completion of the TB screening and diagnostic cascade should be provided at the health facility in high burden settings. In addition, health care workers should be equipped with skills to diagnose TB.

Much as the yield of community screening low, community screening has its role in TB case finding as it not only reaches populations that are disproportionately affected by access barriers but is also an avenue to facilitate behavioral change on early health seeking behavior among patients with presumptive TB. For its yield to improve, general community screening should be discouraged and instead systematic and targeted screening provided to those at highest risk including contacts and people living in TB hotspots and those living in communities with access barriers to health facilities.

## Supporting information

**S1 Data Facility vs community case finding published data.**
(XLS)

## Acknowledgments

CIDRZ Staff: Tumeyo Phiri is acknowledged for his role in processing sputum samples collected under the study; Joel Bwalya is acknowledged for his role in data entry; Kella Siame, Priscilla Chisenga and Mercy Mwale are acknowledged for their role in data collection, the community and facility health workers at George clinic are acknowledged for their active participation in the project.

Ministry of Health staff: Bridget Banda, Martha Tembo, Njobvu Nkumbwizya, Bridget Nchimunya, Charity Mumba, Abigail Chinkondya, Mercy Chansa, Wilfred Njeleka are acknowledged for working closely with the project staff.

## Author Contributions

**Conceptualization:** Monde Muyoyeta.

**Data curation:** Paul Somwe, Jacob Mutale, Kanema Chiyenu, Mwansa Lumpa, Monde Muyoyeta.

**Formal analysis:** Monde Muyoyeta.

**Funding acquisition:** Monde Muyoyeta.

**Investigation:** Mary Kagujje, Lophina Chilukutu, Winfrida Mwanza.

**Methodology:** Monde Muyoyeta.

**Project administration:** Mary Kagujje, Monde Muyoyeta.

**Software:** Jacob Mutale, Kanema Chiyenu, Mwansa Lumpa.

**Supervision:** Mary Kagujje, Lophina Chilukutu, Paul Somwe, Jacob Mutale, Mwansa Lumpa, Winfrida Mwanza, Monde Muyoyeta.

**Validation:** Lophina Chilukutu.

**Writing – original draft:** Mary Kagujje, Paul Somwe, Monde Muyoyeta.

**Writing – review & editing:** Mary Kagujje, Lophina Chilukutu, Paul Somwe, Jacob Mutale, Kanema Chiyenu, Mwansa Lumpa, Winfrida Mwanza, Monde Muyoyeta.

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
