## [Decision Letter · Decision Letter 0]

21 May 2020

PONE-D-20-06815

Active TB case finding in a high burden setting; comparison of community and facility-based strategies in Lusaka, Zambia

PLOS ONE

Dear Dr. Kagujje,

Thank you for submitting your manuscript to PLOS ONE. After careful consideration, we feel that it has merit but does not fully meet PLOS ONE’s publication criteria as it currently stands. Therefore, we invite you to submit a revised version of the manuscript that addresses the points raised during the review process.

We would appreciate receiving your revised manuscript. To enhance the reproducibility of your results, we recommend that if applicable you deposit your laboratory protocols in protocols.io, where a protocol can be assigned its own identifier (DOI) such that it can be cited independently in the future. For instructions see: http://journals.plos.org/plosone/s/submission-guidelines#loc-laboratory-protocols

We look forward to receiving your revised manuscript.

Kind regards,

Frederick Quinn

Academic Editor

PLOS ONE

Journal Requirements:

Reviewers' comments:

Reviewer's Responses to Questions

**Comments to the Author**

1. Is the manuscript technically sound, and do the data support the conclusions?

Reviewer #1: Yes

Reviewer #2: Partly

Reviewer #3: Yes

2. Has the statistical analysis been performed appropriately and rigorously? 

Reviewer #1: N/A

Reviewer #2: No

Reviewer #3: Yes

3. Have the authors made all data underlying the findings in their manuscript fully available?

Reviewer #1: Yes

Reviewer #2: No

Reviewer #3: Yes

4. Is the manuscript presented in an intelligible fashion and written in standard English?

Reviewer #1: Yes

Reviewer #2: Yes

Reviewer #3: Yes

5. Review Comments to the Author

Reviewer #1: Active case finding for tuberculosis (TB) is recommended by the World Health Organization as an approach to find particularly in high-burden settings, that why that manuscript is so relevant for the control of TB in development world.

Line 107 - History of cough, fever, night sweats, weight loss, chest pain and loss of appetite was documented for all patients presenting for TB screening;

The authors should clarify if all the TB symptoms screening were used based WHO's recommended.

The authors should explain how the diagnostic testing followed the standard of care for TB diagnosis in Lusaka. How were all sputum samples dropped off in study clinics and transported to the laboratory or clinic that have a facility like Xpert MTB/RIF testing.

In data analysis, the authors should be expanded and clarified to ensure that readers understand exactly what the researchers studied. Alternatively, they could include another statistical method as Poisson regression to fit the different variables. So that, the tables will be easier to understand.

Reviewer #2: I congrautulate the authors on conducting this piece of work. Indeed any extra-case detected is a couple of leaves saved from

exposure. The authors have carried out an implementation study whose objective appears to be, demonstrating the utility of

internvensions to improve active case finding at the facility and community. However i find it rather difficult to make out

what the actual intervention is, in other words there is need for clarity on what it is that is being done.

On first reading the study appears to have used a massive sample size but actually the samples that end up being tested to

confirm TB are only about 15-16%, this is because, the data has been filtered through a number of steps and it make it hard to figure out what the denominator is, surely the denominator ought to be with samples tested and the conclusions restricted to tested samples.

The extrapolation would be acceptable if the samples that get tested were randomly selected but this is not the case here. So the interpretation here my be somewhat misleading.

Abstract

The abstract is well written but the results reporting could benefit from clarifying the denominators

of each of the proportions reported and these out to be restricted to the samples tested. Only then will

it be possible to assess if the conclusions are appropriate for the results.

Introduction

Line 51, the authors state that 24,929 cases are missed in Zambia, this predominantly occurs in

In peri urban setting and that 50% missed at facilities. Can they expand on why 50% are missed at

facilities, is it a methodological?, awareness?, capacity problem? Ultimately this is what

the study aims to narrow ....as a gap!!

The introduction in general lacks the context within which the study is set, how do i evaluate the

intervention, who has used this interventions in the way you have designed it?, what does WHO recommend,

Zambia's TB incidence is ~ 208 per 100,000 persons per year, So, comparatively how is this approach expected to improve ACF relative to where it has been implemented?... will you approach improve time to identification, cost of extra case identified, or raw numbers of identified?... what is your measure of improvement.. With such a triangulation of information, one could fully understand what you are doing, how it compares elsewhere in terms of benefits and the anticipated improvements in ACF at detection level and cost involved...

Study setting and study population

There is limited information about the study site and population. For example one can not make out

the geographic distribution of the population used in your study, this is critical information as

the prevalence of TB by such sub-locations is likely to be known, it helps one evaluate the appropriateness of the

communities to include and the depth of the followup all of which determine how the scarce resources are apportioned.

Data collection and data management

What is the data availability policy to enhance the provenance of this?

Data analysis

The authors have not explained the statistical methods used, so it makes it hard to understand what analysis was done.

In my opinion there has been limited statistical input to the work

with the samples size used, a number of approaches could have been used to robustly analyse the data

The authors use the work Impact on case finding?, how is impact defined, raw numbers, proportion, cost per sample found?,

lives potentially save frome exposure

I see tables cited in materials and methods, to me this would mean results are being reported in MM

Results

It is shame, with this amount of data, alot of statistical analysis to properly unpick these characteristics

There samples are selected through a number of non-random selection criteria which makes extrapolations difficult

We need to be sure of the denominators used. From your selection procedure, you screened ~ 50%/50% the community and hospital

and the positive cases 91.5% and 8.5%, could you give an idea of what it costs to identify these (0.08*8,348).... actually the denominator should be the

the samples tested. Actually it appears the prevalence is 14.3% among tested at the facility and 3.6 in the community

What is the utility of p-values on line 187, in this case?? are you detecting more than you would expect?

I suspect the best use of this would be to establish that the ACF with and without any in intervention was significantly different

i.e. Q3 2016, Q3 2017... that is hwre the statistical differences would carry more weight!!

On line 194 you introduce, incidence at national level, is this published or is it data that comes from the your database. How can

that be independently verified, or is this peer reviewed data

It would be better if table 4 showed temporal gain rather than a compounded cumulative change

Discussion

I am unsure what the proportions represent, (in reference to the query raised on the denominator),

it makes it rather hard to follow the discussion

Reviewer #3: The very important topic needs only a punctual review to be published. the conclusions could be further elaborated to guide health systems in Africa.the objectives must be clear, the methodology described and the discussion of results must be aligned with the results presented

6. PLOS authors have the option to publish the peer review history of their article (what does this mean?). If published, this will include your full peer review and any attached files.

Reviewer #1: No

Reviewer #2: No

Reviewer #3: Yes: Osvaldo Frederico Inlamea

---

## [Author Response · Author response to Decision Letter 0]

5 Jul 2020

Reviewer 1: 

Comment 1: Line 107-History of cough, fever, night sweats, weight loss, chest pain and loss of appetite was documented for all patients presenting for TB screening;The authors should clarify if all the TB symptoms screening were used based WHO's recommended

Response 1: We have clarified that cough, fever, weight loss and night sweats are the WHO recommended symptoms for TB screening and that loss of appetite and chest pain are recommended by the Zambia TB guidelines.See methods section, line 127-130, Page 7

Comment 2: The authors should explain how the diagnostic testing followed the standard of care for TB diagnosis in Lusaka. How were all sputum samples dropped off in study clinics and transported to the laboratory or clinic that have a facility like Xpert MTB/RIF testing

Response 2: We have clarified that our algorithms are similar to the standard of care in Zambia. See methods section, line 132-134, page 7. The project installed a geneXpert instrument at the study site, George clinic. All the samples, both from the community and the facility, were transported to the laboratory for geneXpert MTB/RIF testing on the same day of collection by community health workers. This has been clarified in the methods section, line 103, page 6 and line 142-143, page 7.

Comment 3:In data analysis, the authors should be expanded and clarified to ensure that readers understand exactly what the researchers studied. Alternatively, they could include another statistical method as Poisson regression to fit the different variables. So that, the tables will be easier to understand

Response 3: Thank you for the feedback. We have made various changes to the data analysis section to make it more clear. See methods section, line 168-189, page 8-9. Additionally, Poisson regression would not be suitable statistics to run on the data as this was a prospective cross sectional study. 

Reviewer 2: 

Comment 1:The abstract is well written but the results reporting could benefit from clarifying the denominators of each of the proportions reported and these out to be restricted to the samples tested. Only then will it be possible to assess if the conclusions are appropriate for the results

Response 1: The denominators have been added to the parts where they were missing. See abstract section, line 41 page 2

Comment 2:Line 51, the authors state that 24,929 cases are missed in Zambia, this predominantly occurs In peri urban setting and that 50% missed at facilities. Can they expand on why 50% are missed at facilities, is it a methodological?, awareness?, capacity problem? Ultimately this is whatthe study aims to narrow ....as a gap!!

Response 2: We have included the various reasons why TB cases are missed. See introduction section, line 59-62, page 4

Comment 3:The introduction in general lacks the context within which the study is set, how do i evaluate the intervention, who has used this interventions in the way you have designed it?, what does WHO recommend. Zambia's TB incidence is ~ 208 per 100,000 persons per year, So, comparatively how is this approach expected to improve ACF relative to where it has been implemented?... will you approach improve time to identification, cost of extra case identified, or raw numbers of identified?... what is your measure of improvement. With such a triangulation of information, one could fully understand what you are doing, how it compares elsewhere in terms of benefits and the anticipated improvements in ACF at detection level and cost involved...

Response 3: The context of the study is provided under methodology section, line 87- 103, page 5-6. We have clarified that our algorithms are WHO recommended and are similar to the standard of care Zambia. See methods section, line 132-134, page 7. Our approach did not measure time to identification but rather measured increase in TB case detection by using a quasi-experimental approach, before and after comparison. The overall case finding approach was measured in terms additional cases detected and change in TB notification rate. This was been clarified in the introduction section, line 81-83, page 5. Facility case finding and community case finding are being compared in terms of yield of bacteriologically confirmed TB diagnosis. See introduction section, line 81-83, page 5. Yield has been defined as bacteriologically confirmed TB cases detected among those who submitted sputum. See methods section line 172-173, page 9. The comment about TB incidence in Zambia is noted. In the methods section, under study setting and study population sub section, we have included the notification rates for Lusaka district as well as the pre- intervention notification for George health facility to better contextualise the ACF relation to notification rate. methodology section, line 94- 98, page 5. We have noted the comment of the cost of each additional case found. The costing aspect of this study will be reported in separate manuscript. 

Comment 4:There is limited information about the study site and population. For example one can not make out the geographic distribution of the population used in your study, this is critical information as the prevalence of TB by such sub-locations is likely to be known, it helps one evaluate the appropriateness of the communities to include and the depth of the followup all of which determine how the scarce resources are apportioned

Response 4: Geographical distribution and location of the population has been included. See methods section, line 87-92, page 5

Comment 5:What is the data availability policy to enhance the provenance of this?

Response 5: The data has been shared under additional information

Comment 6: The authors have not explained the statistical methods used, so it makes it hard to understand what analysis was done. In my opinion there has been limited statistical input to the work with the samples size used, a number of approaches could have been used to robustly analyse the data. The authors use the work Impact on case finding?, how is impact defined, raw numbers, proportion, cost per sample found?,lives potentially save from exposure

Response 6: We have made various changes to the data analysis section in order to make it more clear. See methods section, line 168-189, page 8-9. The overall case finding approach was measured in terms additional cases detected and change in TB notification rate. This was been clarified in the introduction section, 81-83, page 5. Facility case finding and community case finding are being compared in terms of yield. See introduction section 82-83, page 5. Yield has been defined as bacteriologically confirmed TB cases detected among those who submitted sputum. See methods section line 172-173, page 9.

Comment 7:I see tables cited in materials and methods, to me this would mean results are being reported in MM

Response 7: The citations have been removed

Comment 8: It is shame, with this amount of data, alot of statistical analysis to properly unpick these characteristics. There samples are selected through a number of non-random selection criteria which makes extrapolations difficult, We need to be sure of the denominators used. From your selection procedure, you screened ~ 50%/50% the community and hospital and the positive cases 91.5% and 8.5%, could you give an idea of what it costs to identify these (0.08*8,348).... actually the denominator should be the the samples tested. Actually it appears the prevalence is 14.3% among tested at the facility and 3.6 in the community

Comment 8: The analysis of data follows the TB diagnostic cascade. At each level of the cascade, the denominator changes. This is reflected in the flow diagram and has been clarified under the methods section, line 169-173, page 8-9. We have added denominators to the figures in the results section, line 199-249, page 10-12. Information on cost will be provided in a different manuscript

Indeed, the yield is dependent on the samples tested. The overall yield for facility and community case finding case finding were 13.7% and 3.1% respectively. This is shown in the results section, Line 2116, page 11 and line 226 page 11 respectively

Comment 9:What is the utility of p-values on line 187, in this case?? are you detecting more than you would expect?I suspect the best use of this would be to establish that the ACF with and without any in intervention was significantly different i.e. Q3 2016, Q3 2017... that is hwre the statistical differences would carry more weight!!

Response 9: The p-values were originally intended to show that there was a statistically significant difference between the yield for community and facility case finding. To improve the flow of the results section, the content on line 185-187 which include the p-values has since been deleted and the yields put under the respective sections.thanks for the suggestion on use of p-values to show effect of ACF, however, we want to focus this data on the trend in TB notifications

Comment 10:On line 194 you introduce, incidence at national level, is this published or is it data that comes from the your database. How can that be independently verified, or is this peer reviewed data

Response 10: The notification rate is for the facility. We have provided data on George notifications and the population in 2016, 2017 and 2018 so that it is possible to verify the notification rate. A clear table on this is provided in results section, line 255, page 13

Comment 11: It would be better if table 4 showed temporal gain rather than a compounded cumulative change

Response 11: We don’t understand what the reviewer is suggesting. We would like further explanation on this for us to adequately respond.

Comment 12: I am unsure what the proportions represent, (in reference to the query raised on the denominator),it makes it rather hard to follow the discussion

Response 12: Thank you. We have added denominators to the proportions in the discussion section, line 301 page 15

Reviewer 3

Comment 1:The very important topic needs only a punctual review to be published. the conclusions could be further elaborated to guide health systems in Africa.the objectives must be clear, the methodology described and the discussion of results must be aligned with the results presented

Response 1:Thank you so much.We have made some few changes to the conclusion, see conclusion section line 313- 227, Page 15-16. We have clarified the objectives, see introduction section, line 78-83, Page 5. Various changes have been made to the methods and discussion section, please see respective sections. Methods section, line 86-189, page 5-9 and discussion section, line 258-310, page 13-15

PONE-D-20-06815 Reviewer 

Comment 1: Please make it clear what the aim of the study is and how it can be measurable. both in the abstract and in the text 

Response 1: This has been clarified. See abstract section, line 25-29, page 2 and introduction section, line 78-83, page 5

Comment 2: Describe the community, is it around urban area? Urban or rural? where are the health facilities located serving the population? What population do these health facilities serve? is it differentiated in terms of knowledge level, socio-economic status? 

Response 2: This has been clarified under methods section, line 87-92, page 5

Comment 3: Were there differences in the time of sputum collection and sample reception? Those patients were informed about how to collect sputum correctly? 

Response 3: We have clarified that samples were received in the laboratory on the same day that they were collected. See methods section, line 142-143, page 7. All patients were instructed on how to collect sputum samples. See methods section, line 142-143, page 7

Comment 4: Please Clarify what are the criteria for rejection of sputum (sputum not evaluated 2,173

Response 4: Samples were rejected by the laboratory if: i) the specimen was leaking out into biohazard bag, ii) the sputum contained many food particles, iii) the volume was less than <0.5mls and if the sputum contained a lot of blood. See methods section, line 143-145, page 7

Comment 5: Explain in the discussion probable cause of low MTB detection in the sputum (only 506). 

Response 5: The 506 cases were detected from 3528 presumptive TB cases giving a yield of 14.3%. This was higher than the 10% target provided in the Zambia National TB guidelines. See discussion section, line 274-275, page 14

Comment 6: In line 107 until 109 you describe different symptoms for TB including duration of cough but in table 1. You describe statistics of cough and symptoms, please clarify if you include or no the cough with other symptoms. 

Response 6: Cough is included in the other symptoms. Cough has now been removed since it is included in the other symptoms 

Comment 7: Was the information from line 107 to 109 used to describe typical characteristics of TB?

Response 7: We have clarified that cough, fever, weight loss and night sweats are the WHO recommended symptoms for TB screening and that loss of appetite and chest pain are recommended by the Zambia TB guidelines. See methods section, line 127-130, Page 7

Comment 8: improve the methodology and results in the abstract. so you can answer first the objective of the study.

A total of 18,194 individuals were screened through the facility 9,846 (54.1%) and 8,348 (45.9%) were screened through the community. 9,309 (51.2%) were male..... 

Response 8:Both methodology and results section have been revised. See abstract section, line 31-44, page 2

---

## [Decision Letter · Decision Letter 1]

6 Aug 2020

Active TB case finding in a high burden setting; comparison of community and facility-based strategies in Lusaka, Zambia

PONE-D-20-06815R1

Dear Dr.Kagujje,

We’re pleased to inform you that your manuscript has been judged scientifically suitable for publication and will be formally accepted for publication once it meets all outstanding technical requirements.

Kind regards,

Frederick Quinn

Academic Editor

PLOS ONE

Additional Editor Comments (optional):

Reviewers' comments:

Reviewer's Responses to Questions

**Comments to the Author**

1. If the authors have adequately addressed your comments raised in a previous round of review and you feel that this manuscript is now acceptable for publication, you may indicate that here to bypass the “Comments to the Author” section, enter your conflict of interest statement in the “Confidential to Editor” section, and submit your "Accept" recommendation.

Reviewer #1: (No Response)

Reviewer #2: All comments have been addressed

2. Is the manuscript technically sound, and do the data support the conclusions?

Reviewer #1: Yes

Reviewer #2: Yes

3. Has the statistical analysis been performed appropriately and rigorously? 

Reviewer #1: Yes

Reviewer #2: I Don't Know

4. Have the authors made all data underlying the findings in their manuscript fully available?

Reviewer #1: Yes

Reviewer #2: Yes

5. Is the manuscript presented in an intelligible fashion and written in standard English?

Reviewer #1: Yes

Reviewer #2: Yes

6. Review Comments to the Author

Reviewer #1: The authors have clarified the doubts in the first review.Targeting should be applied in TB interventions to improve yield and use resources more efficiently in Lusaka, Zambia.A screening tool should be provided for vulnerable populations to guide health workers to identify presumptive cases.

The authors used a representative number of individuals to screen TB in the community. Data improvement for TB, specifically synchronizing laboratory and TB registers, is another timely recommendation, especially in the current context of TB in Lusaka.

Reviewer #2: The paper is reporting findings from an implementation study, while i think a more robust statistical analysis could be done, i accept that the authors statement that they have another paper in preparation that would capture that. Therefore this would be considered a descriptive paper.

7. PLOS authors have the option to publish the peer review history of their article (what does this mean?). If published, this will include your full peer review and any attached files.

Reviewer #1: No

Reviewer #2: **Yes: **Adrian Muwonge

---

## [Editor Report · Acceptance letter]

19 Aug 2020

PONE-D-20-06815R1 

Active TB case finding in a high burden setting; comparison of community and facility-based strategies in Lusaka, Zambia 

Dear Dr. Kagujje:

I'm pleased to inform you that your manuscript has been deemed suitable for publication in PLOS ONE. Congratulations! Your manuscript is now with our production department. 

Kind regards, 

on behalf of

Dr. Frederick Quinn 

Academic Editor

PLOS ONE